# Analysis of the Circular Economic Production Models and Their Approach in Agriculture and Agricultural Waste Biomass Management

**DOI:** 10.3390/ijerph17249549

**Published:** 2020-12-20

**Authors:** Mónica Duque-Acevedo, Luis Jesús Belmonte-Ureña, Natalia Yakovleva, Francisco Camacho-Ferre

**Affiliations:** 1Department of Agronomy, Research Centre CIAIMBITAL, University of Almería, 04120 Almería, Spain; mda242@ual.es (M.D.-A.); fcamacho@ual.es (F.C.-F.); 2Department of Economy and Business, Research Centre CIAIMBITAL, University of Almería, 04120 Almería, Spain; 3Newcastle University Business School, Newcastle University, London E1 7EZ, UK; Natalia.Yakovleva@newcastle.ac.uk

**Keywords:** circular economic production model, circular economy, bioeconomy, circular bioeconomy, agriculture, agricultural waste biomass

## Abstract

As of now, circular economic production models of the circular economy (CEPMs), which include circular economy, bioeconomy, and circular bioeconomy, are among the main tools characterizing development policies in different countries. During the last five years, policies and strategies regarding CEPMs have promoted and contributed to the development of research on this topic. The evolution and most relevant aspects of the three CEPMs previously mentioned have been analyzed from a sample of 2190 scientific publications obtained from the Scopus database. Bibliometric analysis has been used to evaluate the approach of these models in agriculture and to introduce the ways in which they address the management of agricultural waste biomass (AWB). Results show that the circular economy is the most studied and prioritized model in China and most European countries, with the UK leading the way. Germany leads in topics related to the bioeconomy. The management policies and strategies of the circular bioeconomy are key to promoting research focused on AWB valorization since bioenergy and/or biofuel production continue to be a priority.

## 1. Introduction

The circular economy (CE) and the bioeconomy (BE) have become alternative economic production models (EPMs) that are essential to promote sustainable growth and development [1,2,3]. The main goal of both models is to achieve a synergy between the economy, the environment, and society [4,5,6]. This is why they are key tools in drawing up the policies that need to be implemented to achieve the 17 Sustainable Development Goals (SDGs) of the 2030 Agenda adopted by all United Nations Member States in 2015 [7,8,9,10,11]. This global roadmap raises the need to transform financial, economic, and political systems to improve the quality of life of people around the world, which poses a significant challenge for governments [11,12]. 

The unsustainable use of natural resources and the resulting negative effects on both the environment and the health of human populations have made it vital to take urgent measures to reduce dependence on non-renewable resources [13]. Nevertheless, the global material footprint continues to increase rapidly. The increase in the consumption of natural resources is higher than the improvements in the efficiency to optimize the use of these resources [12]. This is why the transition towards circular economic production models (CEPMs) seems to be the most viable option to abandon the traditional model that led to unsustainable production and consumption levels over many years [14,15,16]. These models, with origins in the ecological economy and the industrial ecology [17,18], promote the efficient use of resources and the utilization of materials with a long life cycle to minimize the generation of residue [1,11,19,20].

Common practices of the circular economy and the bioeconomy not only contribute to the reduction of the impact on natural resources but also represent an opportunity for the creation of employment and local development [21,22,23]. Policies regarding these EPMs include a broad regulatory framework focused on the development of knowledge, research, and innovation regarding the transformation of natural resources, both renewable and non-renewable, to convert them into other products of higher added value [6]. The European Union (EU) defines the CE as an economy that is “restorative or regenerative by intention and design and that aims to maintain the value of products, materials, and resources for as long as possible by returning them into the product cycle at the end of their use while minimizing the generation of waste” [6]. This meaning has been widely promoted by the Ellen MacArthur Foundation [18]. This organization has also proposed that a more holistic approach to water management and circularity can be achieved through the CE by aligning the human water cycle with the natural water cycle. This would require taking action to efficiently reduce water consumption through its reuse, recycling, and replacement. These processes demand technological innovation to reduce the water footprint in industrial processes, agriculture, and other sectors that use considerable amounts of water [24]. Under this circular approach, integrated systems for wastewater treatment and nutrient recovery have been implemented, while combining new technologies and processes that maximize the valorization of high-value by-products for its reuse in agriculture [25,26].

The EU approved in 2015 its action plan, “Closing the Loop-An EU Action Plan for the Circular Economy”, in which water resources are a priority area. This document raises the need for measures to promote the reuse of treated water in agriculture as a strategy to address the pressure on water and contribute to the reduction of the water footprint [27]. This is why the EU issued the Regulation (UE) 2020/741, regarding the minimum requirements for water reuse, on May 25, 2020. Its main goal is to promote the circular economy and guarantee that the water generated is safe for agricultural irrigation [28].

The transformation of non-renewable resources into biomaterials is one of the main innovative aspects in CE programs [6]. The Bioeconomy is specifically related to animals, plants, their derived biomass, and organic waste, among other systems that depend on biological resources [11]. The German Council for Bioeconomics defines the bioeconomy as “the production and utilization of biological resources (including knowledge) to provide products, processes, and services in all sectors of trade and industry within the framework of a sustainable economy” [29].

The European Union believes that a sustainable bioeconomy is the renewable segment of the CE [11,30]. Both models are closely linked and they are complementary, with both aimed at sustainability [6,31]. They share common thematic axes, i.e., biomass, bio-based products, and food waste as areas of an intervention [6]. During the last five years, there has been an insistence on a sustainable and circular bioeconomy [32,33]. In 2018, the EU updated its 2012 strategy, “Innovating for sustainable growth: a bioeconomy for Europe”. Through its new strategy, “A sustainable bioeconomy for Europe: strengthening the connection between economy, society and the Environment - Updated Bioeconomy Strategy” [11], and its action plan, “Bioeconomy: the European way to use our natural resources. Action plan 2018” [10], the EU proposed that sustainability has to be the central axis of the bioeconomy and that this EPM must be circular by definition [10]. This renewed strategy establishes fourteen specific measures which include the facilitation of the development of new biorefineries given their importance for the sustainable processing and transformation of the biomass. It also prioritizes the use of renewable local resources and reactivation of rural areas through the installation of biorefineries in rural environments. The latter is a strategy to improve employment, reduce biomass transportation costs, and to achieve a lower environmental footprint through reducing water and energy consumption [10,11]. 

Since this new management framework was issued, the use of the term “circular bioeconomy” has become common and much research focusing on this concept has been developed [34,35,36,37,38]. Some government policies and/or strategies have taken on this new name for the EPM [39]. One of the primary areas of interest for the bioeconomy and the circular economy is agriculture, due to its elevated production of biomass [11,40]. The utilization of crops for the production of bioproducts, especially bioenergy and biofuel, has increased in recent years [13,30,41,42]. Similarly, agricultural waste represents an important supply in the biological-based economy [43]. Many studies have shown that this type of biomass, in addition to having great energetic potential [23,31,44,45], can be used to manufacture a broad variety of bioproducts, chemicals, and food products [31,37,43,46,47,48,49,50]. In 2013, 17 percent of the dry matter created by primary agricultural waste generated in Europe was used as raw material for the bio-based sectors and bioenergy. In this regard, cereal-producing countries are the largest generators of primary agricultural waste [15,51]. The sustainable use of the resources provided by agricultural biomass is key to prevent risks regarding food insecurity in certain regions [41,42].

Under the approach of the CE, and primarily within the framework of the bioeconomy and the circular bioeconomy, a wide range of opportunities for the recovery and valorization of agricultural residue and sub-products are presented [52]. In recent years, some research has suggested that the increasing number of policies and/or strategies about the CE and the BE has boosted the recovery of agricultural residue and contributed to the improvement of valorization techniques [30,33]. For this reason, this study has three main objectives:To analyze the evolution and main characteristics of the circular economic production models (CEPMs);To evaluate the CEPMs’ approach to agriculture and agricultural waste biomass (AWB) management as the main thematic areas;To examine the interaction between the policies and/or strategies and scientific production in CEPMs.

In this paper, agricultural waste biomass (AWB) refers to crop remains. Several studies have analyzed the scientific research related to the CE, the BE, and the CB (Table 1), mainly through bibliometric analyses. The ten studies presented in Table 1 have been published within the last three years. Their main approach has been the CE. Only one of these papers links the CE to the BE. Among the principal goals of these studies is the evaluation of the concepts, evolution, policies, and strategies related to the CE production models. However, unlike those studies, ours aims to present a joint analysis of the CEPMs of the CE, the BE, and the CB. The main aspects of each trend, as well as the approach of each CEPM, are highlighted. Specifically, we explain how each analyzed model addresses the management of the AWB. Additionally, the complementary evaluation of the policies and/or strategies of the CEPMs allows us to determine their incidence in the evolution of the scientific production on this topic. 

After reviewing the results of the aforementioned studies, there is no evidence that the results of our research are included in any of them. This is why this paper will complement and strengthen the most relevant aspects of this important topic, which has been of special interest in recent years. 

## 2. Materials and Methods 

### 2.1. General Description of the Process

This study was conducted in three stages. The first one consisted of a review and general evaluation of the scientific production on CEPMs. In the second phase, a more specific analysis of the research related to agriculture and agricultural waste biomass was carried out based on the general database. The analyses conducted in the first two stages were done using the bibliometric method. This has been one of the most used methods in recent years for the evaluation of scientific publications [61,62,63] and it is based mainly on three types of indicators. Among them, quantitative indicators focus on productivity, analyzing the number and distribution of publications per year and per journal. Secondly, performance indicators evaluate the quality of the publications through the average number of citations per article, total number of citations, number of authors, and journal impact factor, among others. The statistic indexes that analyze the connections between the authors and the research areas are known as structural indicators [64,65]. 

In this research, quantitative and qualitative bibliometric indicators were used to analyze the evolution of the scientific production on each of the CEPMs and to identify the main characteristics of the selected publications. For the graphic representation of the data, bibliometric network maps were developed with VOSviewer (version 1.6.11). This is free software developed by Dutch Leiden University. It is also a user-friendly tool which allows for the direct processing of bibliometric information from the Scopus database and is ideal for representing and visualizing it through networks [66,67]. Among the bibliometric networks that can be built with this software to enable a comprehensive analysis of scientific activity are citation maps, co-citation, bibliographic coupling, and coauthorship networks [68]. The last stage of the study consisted of the review and analysis of articles on CEPM strategies and/or policies in the countries that were prioritized during the first two stages. Figure 1 represents the main stages of our research. 

### 2.2. Main Stages of the Process

#### 2.2.1. General Analysis of Scientific Production. Objective 1: To Analyze the Evolution and Main Characteristics of the Circular Economic Production Models (CEPMs)

A total of 2190 scientific papers obtained from the Scopus database were extracted and analyzed. There are currently different data sources in the scientific literature (Google Scholar, PubMed, Scopus, and Web of Science) with different approaches in the research fields and tools for data analysis. However, Elsevier’s Scopus is a multidisciplinary database that contains an extensive summary of global research [69]. It is one of the main archives of literature that uses peer review as a method for the validation of scientific research. Scopus has a simple interface and intelligent tools for the analysis and visualization of research and it helps in the process of graphing the data through software such as VOSviewer [30,70]. This is why our analysis was conducted from the information in this database. 

The selected sample only included the final versions of articles [71], reviews, books, and chapters that have been published on this topic through July 2020. According to the search criteria, which are detailed in Figure 1, the period analyzed comprises 18 years. The first paper that appears in the search was published in 2004 and the last one in 2020. Among the variables that were studied are the number of articles per CEPM and their respective citations, main authors, countries and institutions, and distribution of publications by journals. The trend of the approaches regarding the topic and the semantic structure was also analyzed using keyword co-occurrence networks and co-occurrence networks based on textual data from publication titles and abstracts.

#### 2.2.2. Specific Analysis by Subject. Objective 2: To Evaluate the CEPMs’ Approaches in Agriculture and Agricultural Waste Biomass (AWB) Management as Main Thematic Areas

Once the main databases was systematized and the general analyses of the research were done, Scopus database lists, which only included research focused on agriculture and AWB, were created. For the first thematic area, 68 articles were obtained, while 26 others were found relating to the second thematic area. The variables considered for the specific analysis of these studies were the evolution of studies per year and their importance according to the number of citations, main authors and countries, and the evaluation of the approach of the publications based on the analysis of terms and keywords. Figure 1 describes the search criteria selected for each thematic area, the number of studies according to the type of document, and the activities carried out for information processing. 

#### 2.2.3. Search of Policies and/or Strategies on the CEPM in Prioritized Countries. Objective 3: To Examine the Interaction between the Policies and/or Strategies and the Scientific Production on CEPMs

The list of countries with the most articles published on CEPMs was obtained from the databases processed (in general and by thematic area). These countries were prioritized for further analysis of their contributionontributions to policies and strategies on this topic. More than 25 documents were obtained from the websites of the main organizations competent in the subject (European Union, International Advisory Council on Global Bioeconomy (IACGB)). Results obtained from the review of these regulations and management instruments about CEPMs extended the analyses and reflections of the different contributions and progress made by countries regarding this subject. Figure 1 describes the procedure carried out and shows the main search sources. 

## 3. Results and Discussion

### 3.1. Evolution of Scientific Production on CEPMs 

Only one article was published in 2003 and in 2004. Figure 2 shows the trend in the number of yearly publications, which underwent significant variation from 2016. In that year, the number of articles was 144. The year showing the highest number of articles published was 2019 (621). Eighty-five percent of documents were published between 2016 and July 2020. This is a clear indicator of the importance that the CEPMs have gained during the last five years. 

The main CEPM of the analyzed sample is the CE since 68 percent of the articles belong to this model. A total of 652 articles about the BE have been published (30-percent). Finally, under the approach of the CB, 49 articles have been published, which represents 2 percent of the total (Table 2). These results also demonstrate the interest that the scientific community has in these topics. In the first seven months of 2020, 405 articles were published, which equals 65 percent of the total number of articles published in 2019. 

#### 3.1.1. Circular Economy (CE)

The first article on this approach was published in 2004 and was titled “Strategy and Mechanism Study for Promotion of Circular Economy in China” [72]. The last work was titled “Design for Deconstruction Using a Circular Economy Approach: Barriers and Strategies for Improvement”, published 26 July 2020 [73]. During the period analyzed, it is noted that 2017 marked a substantial increase in the number of articles published (171). However, 2019 saw the highest number of publications on this CEPM, with 472 articles (Figure 2). 

#### 3.1.2. Bioeconomy (BE)

The first publication on this CEPM in 2003 was titled “The Bioeconomy and the Forestry Sector: Changing Markets and New Opportunities” [74]. This work was dedicated to the study of the BE in forestry and it proposed that BE would substitute the conventional economic model over the next twenty-five years. It also suggested that this new economy would progressively consolidate the era of biofuels and biochemistry, which are more environmentally friendly processes that can improve the quality of life of many populations. This is a new study that analyzes how to reconcile environmental management and the welfare of stakeholders [75]. The last article, “Evaluating the Impact of Future Global Climate Change and Bioeconomy Scenarios on Ecosystem Services Using a Strategic Forest Management Decision Support System” [76], was published on July 8, 2020.

The increase in the number of publications about this CEPM took place in 2018 (114 articles). However, the year 2019 is when the highest number of articles was published (137 articles). Even though the general trend is a higher number of yearly publications on CE than on BE, there were more articles published on BE between 2011 and 2015. The number of BE articles was double that of CE articles from 2012 through 2013 (Figure 2). 

#### 3.1.3. Circular Bioeconomy (CB)

This CEPM represents only 2 percent of the total articles published. In 2016, the first articles on this topic were published: “A Circular Bioeconomy with Biobased Products from CO_2_ Sequestration” [34] and “Waste Biorefinery Models Towards Sustainable Circular Bioeconomy: Critical Review and Future Perspectives” [35]. The most recent paper was “Refining Biomass Residues for Sustainable Energy and Bio-Products: An Assessment of Technology, its Importance, and Strategic Applications in Circular Bio-Economy” [37]. In 2019, more than twice as many articles were published (15) than in 2018 (7). Up until July 2020, there were 23 articles about this CEPM published (Figure 2), which is 90 percent more than the number of publications in 2016. This shows a trend towards greater use of this concept and further justifies the relevance of the principle of circularity of the model.

The annual evolution of the publications on the different CEPMs (Figure 2) shows that the regulatory and management tools about the CEPMs that have been issued by the European Union have driven the research in this subject. This international organization issued “Innovating for Sustainable Growth: A Bioeconomy for Europe”, its first policy regarding the bioeconomy, in 2012 [22]. From that year through 2015, the highest number of studies focused on the BE was registered (Figure 2). Three years later (2015), the EU adopted the first action plan for the CE, “Closing the Loop-A EU Action Plan for the Circular Economy”, while encouraging the sustained growth of research focused on this economic model from 2016 until 2020. Similarly, the increase in studies focused on CB from 2018 matches the updating of the BE strategy, in which the EU emphasizes the sustainable and circular nature of the BE. 

### 3.2. Main Characteristics of the Published Studies 

#### 3.2.1. Most Cited Articles by CEPM and Productivity of the Authors 

Table 3 summarizes the scientific papers with a higher number of citations during all periods for each of the CEPMs. The total number of citations for the 15 articles that appear in Table 3 (5048) corresponds to 14 percent of the citations of the entire sample. The articles with a higher number of citations fall into the circular economy CEPM. The 2016 article titled “A Review on Circular Economy: The Expected Transition to a Balanced Interplay of Environmental and Economic Systems” [17] is the most cited of the sample (990 citations). 

The publications on the BE ranked second in the most cited articles. The 2015 article “The Role of Biomass and Bioenergy in a Future Bioeconomy: Policies and Facts” [81] is the most cited of this CEPM (307 citations). Among the articles on the CB, the 2016 article titled “Waste Biorefinery Models Towards Sustainable Circular Bioeconomy: Critical Review and Future Perspectives” [35] also has a high record of 259 citations. 

Regarding the most prolific authors, Kean Birch of York University in Canada has the highest number of published articles (13). Birch also has the second-highest number of citations of all authors in Table 4. All of Birch’s articles are related to the BE. His first article was published in 2009 and the most recent was in 2019 [89,90]. His most cited article is “Theorizing the Bioeconomy: Biovalue, Biocapital, Bioeconomics or…What?” [91] in 2013 with 132 citations. 

The author with the most citations is Yong Geng of the China University of Mining and Technology (1619 citations). The main approach of this author’s articles (11) is the CE. With respect to the other authors in Table 4, Professor Gengfue was the first to publish on this topic. His first article, “Developing the Circular Economy in China: Challenges and Opportunities for Achieving ‘Leapfrog Development’” [92], has 264 citations. However, his most cited article, “A Review of the Circular Economy in China: Moving from Rhetoric to Implementation” [93], published in 2013 has 378 citations. His last article, published in 2018, is titled “Integrating Biodiversity Offsets Within Circular Economy Policy in China” [94] and it has been cited 19 times.

The research from these ten authors represents 5 percent of the studies analyzed. Most of these authors’ articles were published between 2014 and 2020, which reinforces the interest that this topic has raised in recent years. According to the classification of the articles by CEPM, a primary focus on the BE is observed, although some authors have published articles on the CE and the BE (Pagliaro, M., Zabaniotou, A., Blumberga, D., Ciriminna, R.). On the other hand, Toppinen, A. and Zabaniotou, A. are the only ones who have targeted some of their research on the CB axis. In the particular case of Zabaniotou, A., 12 articles were published, covering each of the three CEPMs analyzed.

#### 3.2.2. Main Countries and Institutions 

Figure 3 represents the ten countries with the largest scientific production. Contributions from these countries reach a total of 1752 articles that represent 91 percent of all analyzed articles (Table 5). Sixty-nine percent of these studies are related to the CE, 29 percent to the BE, and only 2 percent to the CB. Seven of the ten countries identified are European. Of those, the United Kingdom (UK) carried out the most research, with 284 articles, representing 13 percent of the total. Seventy-nine percent of the United Kingdom studies focus on the CE, 20 percent correspond to research on the BE, and 1 percent target the CB. Italy has the second most published articles, with 251, which represent 11 percent of the sample. Seventy-seven percent of these articles fall into the CE CEPM, 21 percent into the BE, and 2 percent into the CB.

China has the highest percentage of published articles about the CE (91 percent). This is consistent with the results of other research [5]. Only 8 percent of the research conducted in this country is related to the BE and 1 percent to the CB. The main thematic axis of the research in all countries, except Germany and Finland, is the CE. In the case of Germany, 56 percent of all published articles are focused on the BE, 43 percent on the CE, and 1 percent on the CB. These numbers justify Germany’s strategy to revitalize and strengthen their centers for the research and innovation of products, processes, and bio-based services [108].

Fifty percent of the articles published by Finland are focused on the BE, 48 percent on the CE, and 4 percent on the CB. This ranks Finland second in the number of published studies in this last CEPM. In this group of countries, Sweden leads with the highest number of articles focused on the CB with 97 (6). All countries except for France have published articles focused on this CEPM (Table 5), although at a very low percentage.

It is important to highlight that almost all of these countries have specific strategies on the CE and the BE. Some of these management tools have also been complemented with action plans. As shown in Table 5, many of these strategies have been adopted over the last five years. Most of them have been recently updated by different countries showing governmental interest in the continuation of strengthening their policies on the CE, the BE, and the CB. Other countries, such as the United Kingdom, presented their first strategies for the bioeconomy in 2018.

Among the specific strategies recently adopted is Italy’s strategy on BE in 2019 [113], which corresponds to a previous version from 2017 [29,135]. Germany’s national strategy on BE, adopted in January 2020 [117], updated a 2013 version of the same strategy. This country has always been a pioneer in policies related to the BE, having published its first strategy, “National Research Strategy Bio Economy 2030—Our Route Towards a Bio-Based Economy”, in 2010 [108]. In addition, the German government created the German Bioeconomy Council in 2009, with representatives from industry, society, and science, to establish an important advisory body whose main objective is to promote the development of a sustainable bioeconomy in Germany and around the world [136].

In addition to these recent strategies, several countries have developed a broad regulatory and tactical framework with specific programs regarding the production, use, and valorization of biological resources under the same approach of the CEPMs under analysis in this research [29]. The most recent strategies in the UK include “Synthetic Biology Strategy Plan: Biodesign for the Bioeconomy” (2016) [137] and “Building a High Value Bioeconomy: Opportunities From Waste” (2015) [138]. The latter pays special attention to the management of agricultural by-products and residue as raw material for the bioeconomy. An older but particularly important document for the UK strategy on CE and BE is “UK Biomass Strategy” (2007). This is a strategy focused on the bioeconomy through the sustainable use of biomass for the production of fuel and renewable materials [41].

China was the first country to develop a regulatory framework on CE, which is contained in the document “Circular Economy Promotion Law of the People’s Republic of China” (2009) [120]. Among other goals, this regulatory instrument prioritized the development of ecological agriculture. It also considered the possibility of adopting advanced technology for the integral use of agricultural residues, such as straw, among other agricultural by-products [120]. This country has also been promoting policies for the development of the biological industry since 2009 [30].

China had already started to prioritize and promote research and development in the field of the CE before 2009. Its first initiative was presented in the 10th Five-Year Plan for the period 2001–2005 [56]. Since then, the update of these national strategies has included aspects linked to the CE and the BE, such as synthetic biology, new biotechnology, and high-tech industrial innovation [29,118,119,139]. Some studies seem to date China’s interest in the CE to the 90s [56,140].

Spain, France, and Sweden are the countries with the most recent strategies on CE [125,132,134]. Each country’s national strategies are usually complemented with local or regional strategies, which applies to all countries in Table 5. In Spain, the Andalusian regional government adopted the “Andalusian Circular Bioeconomy Strategy” in 2018 [141]. Madrid, Extremadura, Catalonia, and Aragon also have their strategies on EC [142]. Finland is another country with a substantial number of local and regional strategies [143].

The United States has had a renewed strategy on bioeconomy since 2019. Until then, the 2012 strategy, “National Bioeconomy Blueprint”, was in effect [82,144]. The European Circular Economy Stakeholder Platform, launched as a joint initiative by the European Commission and the European Economic and Social Committee (EESC) in March 2017, registered 39 national, regional, and local strategies adopted by public authorities of the member states [143]. All of this points to a global trend towards increased policies on CEPM as already indicated by the German Bioeconomy Council in its 2018 report [29,145].

Table 6 shows the most prolific institutions during the period analyzed. The ten institutions in this table represent 12 percent of the total of published articles. It is notable that each of these institutions is European, with the exception of the Chinese Academy of Sciences. Seven of them are universities and three are public research centers. Fifty-eight percent of studies about CEPMs focus on the CE, 41 percent on the BE, and 4 percent on the CB.

Individually, the Bucharest University of Economic Studies is the institution with the highest number of publications (34). The contributions of this Romanian university, which add up to 116 citations, represent 2 percent of the total. Its most cited article is “Social Responsibility, an Essential Strategic Option for a Sustainable Development in the Field of Bio-Economy”, [146] published in 2019. In general, the studies of this institution are oriented towards the CE (23 articles) and the BE (11 articles) and, to date, there are no publications related to the CB. Italy and the Netherlands have more institutions in the top ten represented in Table 6. The contributions of their universities amount to 49 and 54 articles, respectively (Table 6).

Regarding the importance of the publications, the articles of the Delft University of Technology (29) have a higher number of citations (2019). Among them, its most cited article is the 2017 study titled “The Circular Economy–A New Sustainability Paradigm?”, which has amassed 809 citations [77]. Amongst the public research centers with the most contributions are the European Commission Joint Research Centre (JRC), with 25 publications that have accumulated 725 citations. Seventy-two percent of these publications fall into the bioeconomy CEPM. This is the second highest ranking institution in terms of the number of articles about the BE, after the Universität Hohenheim. It is important to highlight that all the publications of this German University are focused on the BE. This noticeable interest of Europe in the BE has also been observed in other research [5].

The 2015 article titled “The Role of Biomass and Bioenergy in a Future Bioeconomy: Policies and Facts” [81], affiliated to the Joint Research Center, is among the five most cited articles about the bioeconomy CEPM, accounting for 307 citations (Table 3). This research center provides scientific advice to the European Commission in developing policies that may affect all the Member States of the European Union. It should be noted that the European Commission is one of the main entities that fund the scientific production analyzed in this study (2190 articles), with 3 percent of the total. Second in the ranking of funding entities is the National Natural Science Foundation of China with 49 publications, 2 percent of the total. The European Regional Development Fund (ERDF), whose main objective is to strengthen socioeconomic cohesion in the EU, and the Horizon 2020 Framework Programme, the main EU program in research and innovation, are also included in the list of the top five sources of funding of articles in our sample. The European Union has funded 7 percent of the articles in the sample through its different programs and institutions. This fact highlights the importance that European countries place on this type of research.

#### 3.2.3. Distribution of the Publications by Journal

Table 7 shows the journals with the largest number of published articles in the sample. Specifically, the ten journals in this table account for 30 percent of the scientific production that has been analyzed. According to the 2019 SCImago Journal Rank (SJR) indicators, 70 percent of these journals fall into the first quartile (Q1), 30 percent into the second quartile (Q2), and 10 percent into the third quartile (Q3). The main thematic axis of these publications has been the EC, which is present in 74 percent of the articles. Articles written about the BE are in second place, accounting for 23 percent of the sample, and articles on CB account for 3 percent of the total.

In the ranking of the most important journals according to the number of published articles, the *Journal of Cleaner Production* leads, with 222 articles, 10 percent of the total. This journal also holds the largest number of citations (9650). Its 2016 article, “A Review on Circular Economy: The Expected Transition to a Balanced Interplay of Environmental and Economic Systems” [17], is the most cited article of all publications analyzed, with 990 citations (see Table 3). The main focus of this journal is on the environment and sustainability [150].

In second place, the journal *Sustainability* accounts for 187 articles, 9 percent of the total, and 2946 citations. Its 2016 article titled “Designing the Business Models for Circular Economy—Towards the Conceptual Framework” [151] is the most cited from this journal in the analyzed sample (273 citations). This journal also focuses primarily on research on sustainability and sustainable development [152]. The journal *Bioresource Technology* accounts for 32 publications and 915 citations. This journal has the largest number of studies in the framework of the CB, with 44 percent of its articles on this topic. Its 2016 article titled “Waste Biorefinery Models Towards Sustainable Circular Bioeconomy: Critical Review and Future Perspectives” [35] is the most cited (262 citations). As shown in Table 7, the publications of the journals *Resources Conservation and Recycling* (87 articles) and *Waste Management* (25 articles) are focused on the CE CEPM. The opposite is the case for the journal *Biofuels Bioproducts and Biorefining*, which, with 22 publications, has only one article in the field of the EC and the rest (21) are oriented towards the BE and the CB.

### 3.3. Analysis of Keywords: General and by CEPM

Only the keywords defined by the authors in their articles were used for this analysis. Thus, the terms indexed by the Scopus database managers (indexed keywords) have not been considered. This decision is due to the generic nature of some keywords which the database managers have used to catalog the articles. Some examples of these types of keywords are “article”, “human”, or “priority journal”, which are not related to the specific research. In addition, the terms used in the general search that led to this research were not included among the keywords listed in Table 8. In this regard, the 20 keywords shown in Table 8 appear in 46 percent of the articles in our sample. Specifically, the term “sustainability” is the main keyword used and it appears in 10 percent of the articles. The term “recycling” is in second place and it is found in 107 articles, which is 5 percent of the total. Finally, the term “sustainable development” appears in 4 percent of publications on our sample.

The word co-occurrence networks show the main terms associated with each CEPM (Figure 4 and Figure 5). In this regard, it has to be noted that the keywords related to the CB thematic axis have been analyzed together with the ones that come from the BE, given that they are the same ones for both models. The terms “sustainability” and “sustainable development” have the same relevance in both the CE and the BE. Both concepts have been a priority in political agendas on development for decades on a global level. As indicated in other studies, the prioritization of the terms “sustainability” and “sustainable development” in both models ratifies the connection between the CE and the CB as well as their common goal of harmonizing the economic, environmental, and social objectives. These two models are allies for sustainability [1,5,6,31].

Keyword nets are useful to identify the emphasis that each CEPM places on certain aspects or research lines. For example, the CE model highlights the importance of recycling, residue, residue management, resource efficiency, and the evaluation of the life cycle. This is consistent with the goals of the majority of policies and/or strategies about the CE, which are reducing residue, increased recycling, and the improvement of resource efficiency during their life cycle [1,19,20].

In the CE keyword net (Figure 4), China is another relevant term. This makes sense considering that this is one of the countries with the largest amount of publications about the CE (Figure 3). In addition, 4 percent of publications are about this country. Specifically, there are 85 articles with different purposes. In some cases, they are revisions of the policies about the CE in different Chinese regions. In other cases, they are evaluations of the development and/or progress in the implemented strategies. There are also articles that carry out a comparative analysis of Chinese and other national policies, as well as of the main challenges, opportunities, and laws in the industrial sector under the approach of the CEPMs. Almost all keywords that appear in the Chinese sample (85) focus on the CE. Only four of them are related to the BE [158,159,160,161].

On the other hand, the co-occurrence keyword net about the BE prioritizes “biomass”, “biotechnology”, “bioenergy”, “innovation”, and “biofuels” (Figure 5). Germany is one of the countries that stands out in this net given that it accounts for the most publications about the BE (Figure 3). One percent of all publications are about this country. The 23 German articles, 65 percent of which are about the BE and 35 percent about the CE, focus on the study of the current situation, development, and future perspectives of these CEPMs in this country. Some are also comparative research studies that evaluate the German models in comparison with models of other countries. The European Union is prioritized in this net, which makes sense considering that the European Commission Joint Research Centre has the largest number of studies on the BE, totaling 72 percent of its publications (Table 6). Other terms of special interest in this research analysis, such as “agriculture”, have been considered in both nets.

Finally, the net about the CE CEPM incorporates the term “bioeconomy” and the word net of axis BE includes the term “circular economy”, which indicates that both models are integrated and/or complementary [6,31].

The supplementary analysis of the terms included in the titles of the 2190 documents of our sample allows the identification of the focus of the studies. Thus, 40 percent of the articles are linked to the terms shown in Figure 6. Thirteen percent are mainly focused on residue, specifically its use, valorization, bioconversion, and management. Different types of residues are also studied, such as food waste, plastic waste, waste-to-chemicals, urban waste, organic residues, agricultural waste, and forestry residues, among others. This is why management is the second most frequent term at 6 percent.

Five percent of the scientific production analyzes the importance of the transition towards the EC, the BE, and/or the CB. Other articles (5 percent) analyze the approach and perspectives of the CEPMs. The policies, strategies, and other tools linked to these CEPMs are also analyzed. A small percentage of articles show an interest in reseach about the relevance of innovation in the CEPMs (4 percent). Finally, 3 percent of the published articles are case studies that specifically analyze the implementation of the CEPMs in certain countries and regions, as well as the development of specific actions.

### 3.4. Specific Analysis of Research Related to Agriculture

#### 3.4.1. Evolution of the Publications and the Main Articles

Sixty-eight percent of the 2190 documents analyzed are related to agriculture, which represents 3 percent of the sample (Figure 7). The first publications on this approach appeared in 2000 and are titled “European Quality Agriculture as an Alternative Bio-Economy” [162] and “The New Bioeconomy and the Future of Agriculture” [163]. The last article included in our sample was published on July 14, 2020 and is titled “Are Agri-food Systems Really Switching to a Circular Economy Model? Implications for European Research and Innovation Policy” [164]. Fifty-seven percent of the articles are oriented towards the bioeconomy CEPM, 37 percent are in the CE axis, and 6 percent in the CB. The number of published articles about agriculture has increased since 2016. The years that account for most publications are 2018 and 2019, with 16 and 11 articles, respectively.

Table 9 shows the most cited articles on this topic. The article with the most citations (87) is “Divergent Paradigms of European Agro-Food Innovation: The Knowledge-Based Bio-Economy (KBBE) as an R&D Agenda” [165], published in 2013. As it appears in the table, 60 percent of these articles are framed in the BE CEPM and 40 percent in the CE.

#### 3.4.2. Main Characteristics of the Publications

Professor Levidow of The Open University and Professor Viaggi of the Alma Mater Studiorum Università di Bologna are the authors with the most studies on this topic, with three articles each. These two institutions have the most publications, with four each. Italy leads in the number of articles, with 13, followed by Spain with 10 articles and then Germany with nine articles. It should be noted that the German region of Weser-Ems is one of the most developed and efficient agricultural areas in the world [116].

Figure 8 shows the keyword net corresponding to articles about agriculture. These articles highlight the importance of agricultural waste biomass as the main raw material, especially in the bioeconomy. In fact, bioenergy and biofuels result from AWB recovery. Sustainable development continues to be one of the featured topics in most of these publications and the importance of sustainable agriculture is also highlighted. “Biotechnology” and “agricultural residue” are also relevant terms in this net. Eighteen of the 68 articles related to agriculture are focused on AWB management, which makes up 26 percent of the sample.

An analysis of the semantic structure can be conducted from the co-occurrence net based on the terms appearing in the title and abstract of each of the documents related to agriculture. This allows the main approach of the articles to be identified (Figure 9). According to the number of occurrences, the most relevant term is “production”. Twelve percent of the articles on agriculture emphasize food production systems and their transition towards circular models. Other articles lean towards the use of waste and residue to develop new products, compost, bioenergy, and biofuels, among others, from waste and residue. This is why this term is among the most relevant ones. Twenty-four percent of the studies focus on the use and valorization of agricultural waste.

The term “development” ranks second in the number of occurrences. Seven percent of the research highlights the contribution of agriculture to sustainable development and the importance of implementing CEPMs to achieve a more circular economy. The term “study”, with 23 occurrences, shows the articles that focus on the development of case study research based on the perspectives of the CE, the BE, and/or the CB models in specific regions and sectors, such as agro-industry. This is why “food” is also one of the most relevant terms. Twenty-one percent of the articles are related to the agri-food industry and its transition towards CEPMs. These studies analyze the ramifications and main challenges of this sector, providing specific data from agri-food industries and companies. Six percent of articles present biomass as an indispensable resource for the bioeconomy. These studies prioritize the management and valorization of AWB in the framework of the CEPMs.

### 3.5. Specific Analysis of the Research Related to Agricultural Waste Biomass (AWB)

#### 3.5.1. Main Characteristics of the Publications

Table 10 presents 26 articles whose main focus is AWB. These articles account for 1 percent of the sample. Thirty-eight percent of the articles were published between January and July 2020. Three more articles were published in this period than in 2019 (8 articles). Eighty-five percent of these studies have been published during the last three years (2018, 2019, and the first seven months of 2020). Regarding this topic, 62 percent of the articles are geared towards the BE and the CB, while 38 percent pertain to the CE.

The first article was published in 2015 and is titled “Fruit Waste Streams in South Africa and Their Potential Role in Developing a Bio-Economy” [173]. The most recent was published in July 2020 and it is titled “Refining Biomass Residues for Sustainable Energy and Bio-Products: An Assessment of Technology, Its Importance, and Strategic Applications in Circular Bio-Economy” [37]. The most cited article (29) was published in 2019 and is titled “A Spatial Approach to Bioeconomy: Quantifying the Residual Biomass Potential in the EU-27” [43].

The countries with the largest number of publications are Germany (7), Italy (4), and Spain (4). These countries were already among the most prolific (Figure 3). In addition, these are the countries with the most publications on agriculture in the framework of the CEPMs. They are also characterized by being among the seven Member States of the EU with the highest potential for AWB [43,51]. For example, the Spanish province of Jaen is one of the territories with the highest production of AWB from the olive oil industry [43].

#### 3.5.2. Analysis of Keywords and the Semantic Structure of the Research

By observing the net developed from the keyword co-occurrence analysis (Figure 10), it can be noted that the term “bioenergy” is the most relevant. In fact, 19 percent of the articles appearing focus on the use of AWB to produce renewable energy [31,37,44,45,174] (Figure 11). The terms “fertilizers” and “manures” also stand out in this net, as proven by the fact that 10 percent of the articles are about the production of fertilizers from agro-industrial by-products and their use to improve the characteristics of the cultivation soil [175,176]. Eight percent of the research is related to the use of AWB for the production of biofuel [177,178]. For this reason, “biofuel” and “biofuels” are the most relevant terms in this keyword net.

Another relevant term is “waste management” (Figure 10). Twelve percent of the studies are linked to valorization [47,177,179,180] and AWB management [7,44,45] under the approach of the CEPMs (Figure 11). From the analyzed sample, the main valorized residues are tomato leaf waste, rice industrial by-products, banana residue, fruit, and straw remains. Wheat, corn, barley, and rapeseed straw have been identified as some of the main sources of bioeconomic potential due to their high concentration of lignocellulose (80%) [43,46].

Biomaterials such as fiber and ceramic have been obtained from the residue previously described. In addition, it is also possible to obtain other chemical compounds and food products, such as secondary metabolites and mushrooms of high nutritional value (Figure 11). Anaerobic digestion and co-digestion [45,181], fermentation, pyrolysis, and gasification [44,45,182] stand out among the main techniques, processes, and technologies to improve the transformation of AWB (Figure 10). These last two processes are commonly used for the production of biogas and biofuels [52]. Some research has specifically evaluated and quantified the amount of AWB generated in the EU, as well as its potential use and future alternatives [15,43,46]. These studies highlight the importance of AWB as an essential raw material for the development of the BE in Europe [43].

## 4. Conclusions

The CEPMs related to the circular economy, the bioeconomy, and the circular bioeconomy have had an important role in worldwide research during the last five years. In addition, the political agendas of most governments in economically developed countries have continuously prioritized these concepts for almost two decades. In this regard, the 2015 Agenda 2030 has been the driving force behind policies and studies related to these lines of research all over the world. For this reason, these CEPMS have become a planning tool for sustainability and sustainable development.

The increase in research on this topic is associated with the many national, regional, and local CEPM policies and strategies, whether newly created or updates to those already existing. The main reason for this is the fact that these CEPMs encourage the development of knowledge, research, and innovation. For example, European Union policies, programs, and strategies about the CEPMs have been the main funding source of the research analyzed in this study.

The UK has demonstrated the largest amount of scientific production about CEPMs. The main approach carried out in this country is the circular economy. Generally speaking, this model has been the most studied. Sixty-eight percent of the scientific production analyzed focuses on the circular economy. This is why this model is known for the promotion of recycling, residue management, resource efficiency, and the evaluation of the life cycle. In addition, this model has been a reference mainly for European countries. In fact, much of the scientific production about the circular economy comes from European countries, specifically Italy, Spain, and the Netherlands. In almost all cases, these countries have detailed, up-to-date circular economy and bioeconomy strategies. However, China has the most publications on circular economy and its policies and strategies about the circular economy have been the subject of constant analysis. One of the authors with the most publications and the largest number of citations is Chinese, which makes China a point of reference regarding the circular economy.

The bioeconomy ranks second in the number of published articles, with 30 percent of the total. The first publication about CEPMs was about the bioeconomy. In addition, this model appeared before the circular economy in European policies. The country that has contributed the most to bioeconomy research is Germany. In addition to a significant amount of research about this CEPM, Germany has led the creation of management tools and regulations about the bioeconomy. These policies are committed to the research and innovation of biologically based products, processes, and services. The European Union has insisted on the circular nature of the bioeconomy in recent years. For this reason, the circular bioeconomy began to emerge in 2009 as a specific model.

In general, residue is an essential supply for the CEPMs analyzed. In this framework, the main raw material for the bioeconomy is organic waste. This is why agriculture is one of the priority sectors in this model. Thus, a stronger interest in the contributions of this sector to the bio-based model has been noted since 2016. In addition to producing a high amount of primary and residual biomass, these agricultural residues have great potential for obtaining products of high added value. AWB is becoming more and more important in the bioeconomy, which demands more research and strategies for its valorization. Despite the fact that an important percentage of AWB is already being used to produce biomaterials, chemical compounds, and food products, the valorization of this AWB for the production of bioenergy and/or biofuel is becoming more important. This raises the need to continue promoting other types of use for this important resource.

## Figures and Tables

**Figure 1 ijerph-17-09549-f001:**
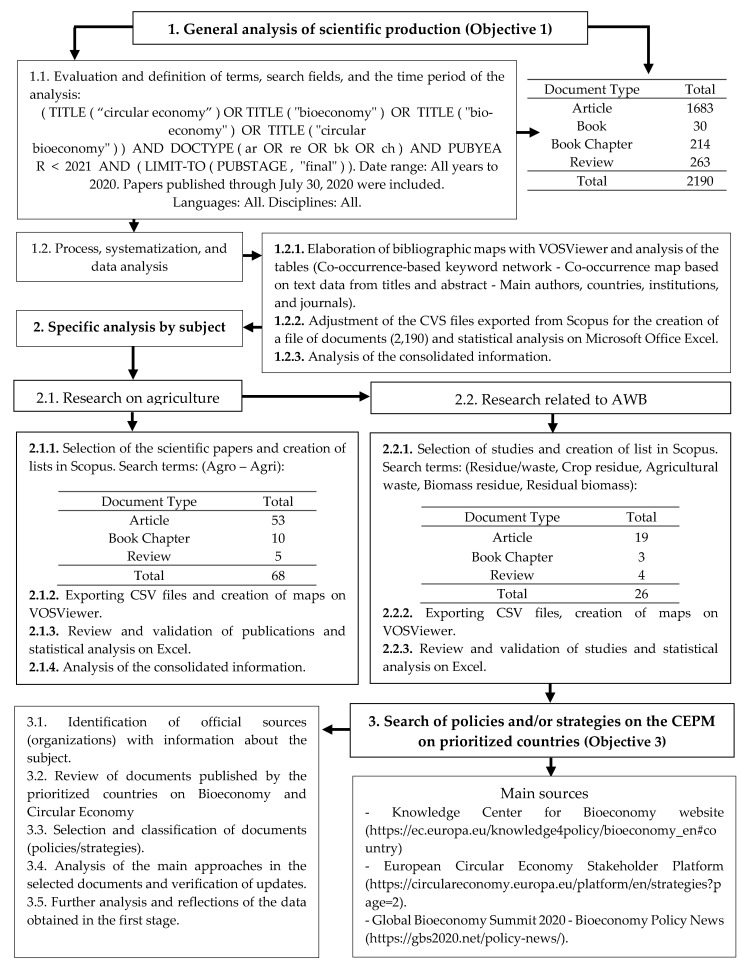
Main stages of methodology used.

**Figure 2 ijerph-17-09549-f002:**
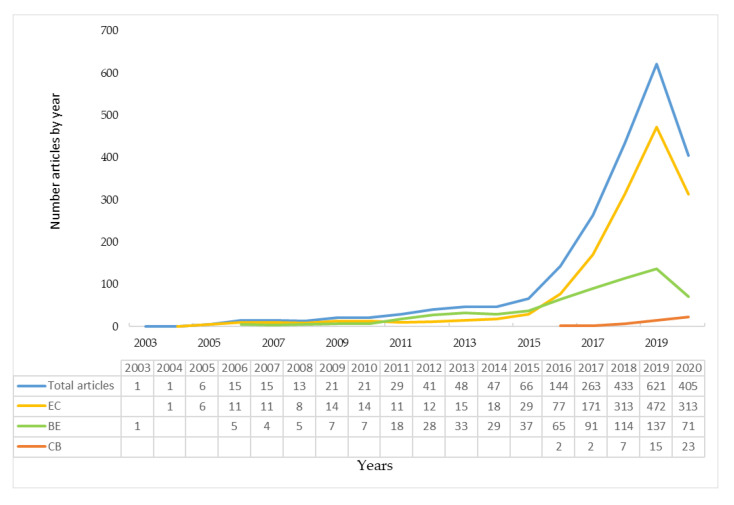
Evolution of scientific production by CEPM.

**Figure 3 ijerph-17-09549-f003:**
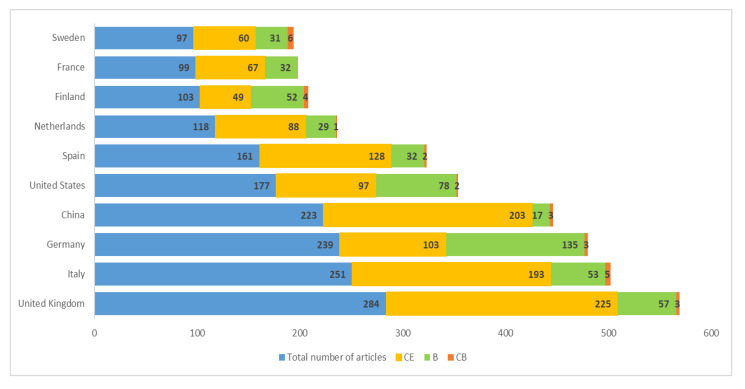
Countries with most published articles by CEPM.

**Figure 4 ijerph-17-09549-f004:**
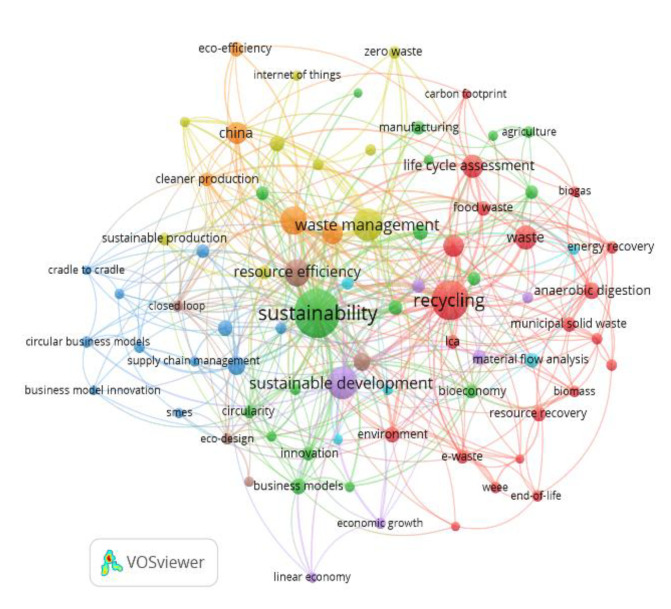
Keyword net based on the co-occurrence for CE.

**Figure 5 ijerph-17-09549-f005:**
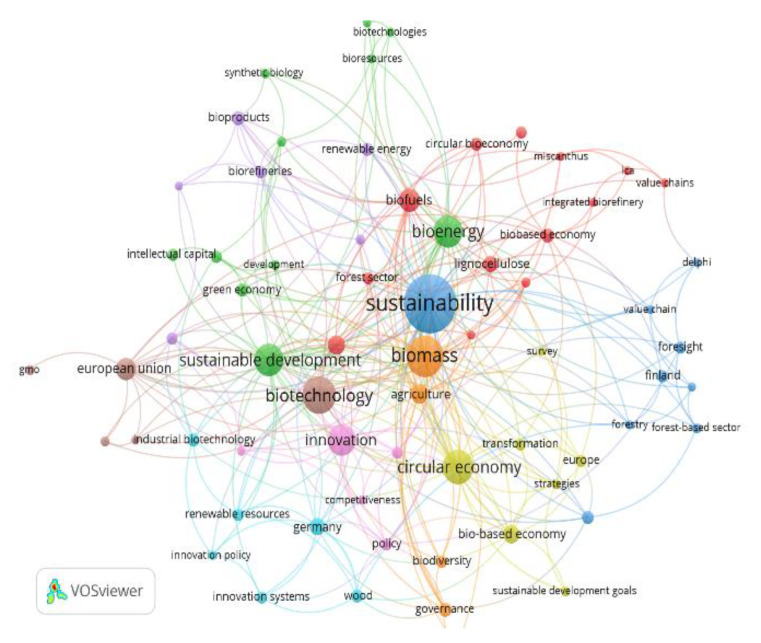
Keyword net based on the co-occurrence for BE and CB.

**Figure 6 ijerph-17-09549-f006:**
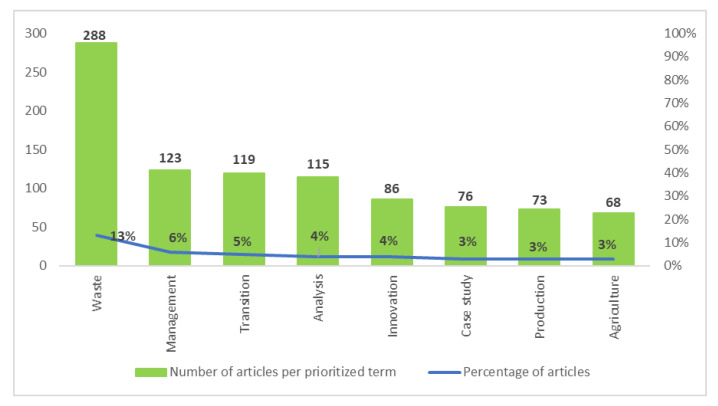
Most relevant terms from the net based on the titles of the documents.

**Figure 7 ijerph-17-09549-f007:**
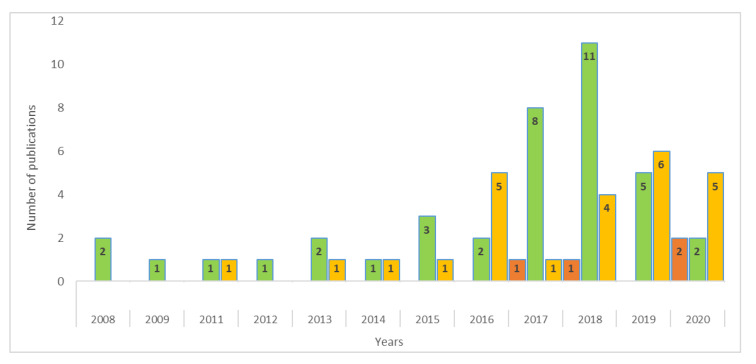
Publications about agriculture by CEPM from 2008 through July 2020.

**Figure 8 ijerph-17-09549-f008:**
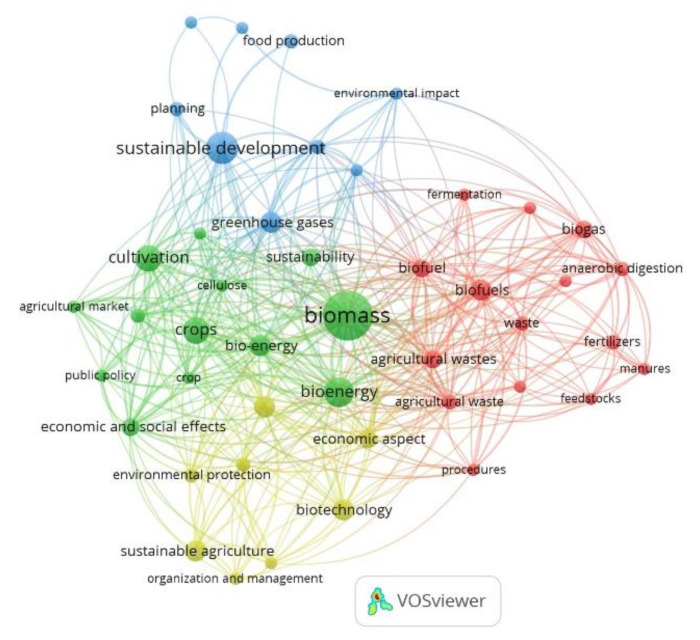
Keyword net for articles related to agriculture, 2008 through July 2020.

**Figure 9 ijerph-17-09549-f009:**
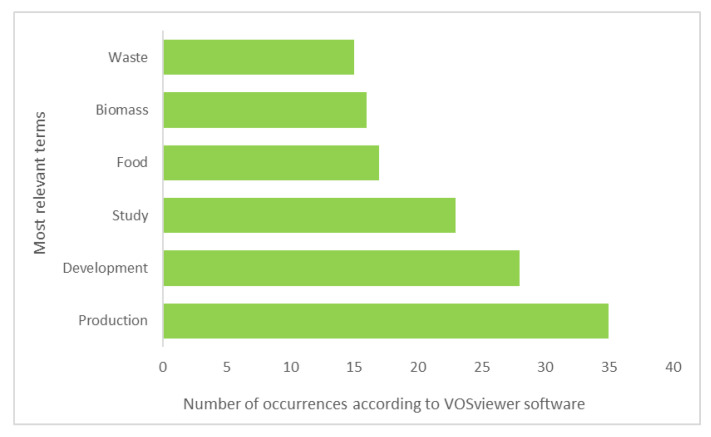
Most relevant terms from the network based on document titles and abstracts.

**Figure 10 ijerph-17-09549-f010:**
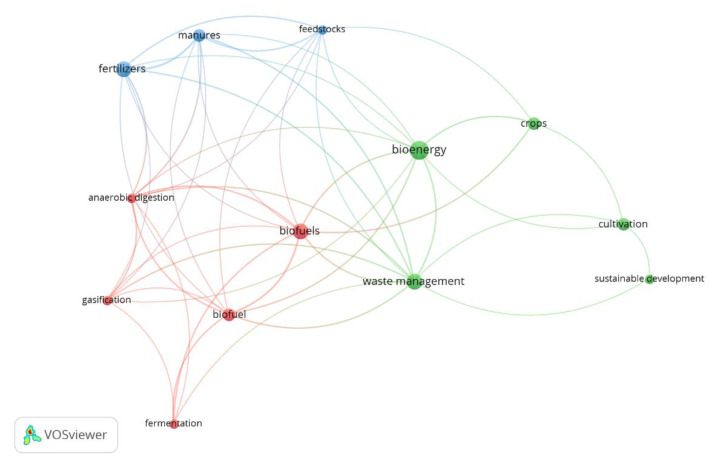
Keyword net from AWB research.

**Figure 11 ijerph-17-09549-f011:**
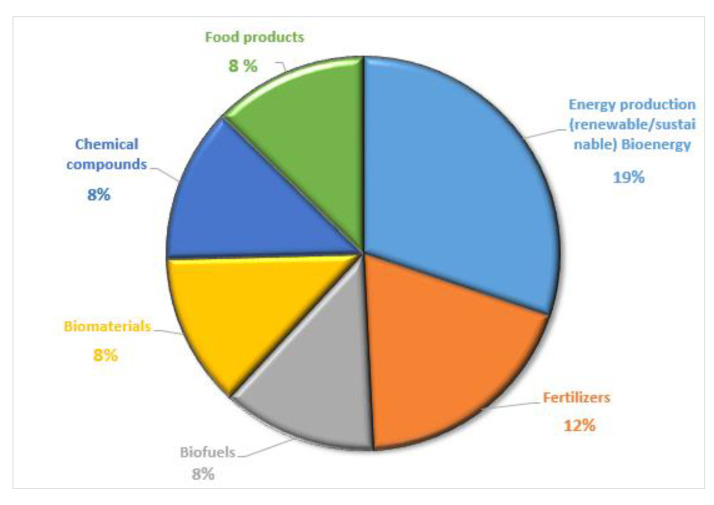
Main products obtained from AWB (percentage of items).

**Table 1 ijerph-17-09549-t001:** Previous studies on Circular Economic Production Models (CEPMs) from 2017 through July 2020.

Year	Name of the Study	Reference
2017	La producción científica española en el ámbito de la bioeconomía. 2005–2014	[53]
2017	Scientific literature analysis on big data and internet of things applications on circular economy: a bibliometric study.	[54]
2017	Green, circular, bio economy: A comparative analysis of sustainability avenues	[5]
2018	Worldwide research on circular economy and environment: A bibliometric analysis.	[55]
2018	Bibliometric and review of the research on circular economy through the evolution of Chinese public policy.	[56]
2018	A definition of bioeconomy through the bibliometric networks of the scientific literature.	[57]
2018	Circular economy scientific knowledge in the European Union and China: A bibliometric, network and survey analysis (2006–2016).	[58]
2020	A literature review on forest bioeconomy with a bibliometric network analysis.	[59]
2020	Effects of Circular Economy Policies on the Environment and Sustainable Growth: Worldwide Research.	[60]
2020	The circular bioeconomy: Its elements and role in European bioeconomy clusters.	[33]

**Table 2 ijerph-17-09549-t002:** Summary of documents by CEPM from 2003 through July 2020.

Document Type	%	Documents by CEPM		
		CE	BE	CB
Article	77%	1232	431	25
Book	1%	10	20	0
Book Chapter	10%	109	101	5
Review	12%	145	100	19
Total	100%	1496	652	49
		68%	30%	2%

CE: Circular Economy; BE: Bioeconomy; CB: Circular Bioeconomy.

**Table 3 ijerph-17-09549-t003:** Articles with higher number of citations by CEPM from 2003 through July 2020.

CEPM	A	AN	TC	Y	J	R
Circular Economy	A review on circular economy: The expected transition to a balanced interplay of environmental and economic systems	Ghisellini, P., Cialani, C., Ulgiati, S.	990	2016	*Journal of Cleaner Production*	[17]
	The Circular Economy—A new sustainability paradigm?	Geissdoerfer, M., Savaget, P., Bocken, N.M.P., Hultink, E.J.	794	2017	*Journal of Cleaner Production*	[77]
	Product services for a resource-efficient and circular economy—A review	Tukker, A.	586	2015	*Journal of Cleaner Production*	[78]
	Conceptualizing the circular economy: An analysis of 114 definitions	Kirchherr, J., Reike, D., Hekkert, M.	572	2017	*Resources*, *Conservation and Recycling*	[79]
	Towards circular economy implementation: A comprehensive review in context of manufacturing industry	Lieder, M., Rashid, A.	507	2016	*Journal of Cleaner Production*	[80]
Bioeconomy	The role of biomass and bioenergy in a future bioeconomy: Policies and facts	Scarlat, N., Dallemand, J.-F., Monforti-Ferrario, F., Nita, V.	307	2015	*Environmental Development*	[81]
	The Bioeconomy in Europe: An Overview	McCormick, K., Kautto, N.	263	2013	*Sustainability (Switzerland)*	[82]
	Strategies and policies for the bioeconomy and bio-based economy: An analysis of official national approaches	Staffas, L., Gustavsson, M., McCormick, K.	183	2013	*Sustainability (Switzerland)*	[83]
	Limonene: A versatile chemical of the bioeconomy	Ciriminna, R., LomeliRodriguez, M., DemmaCarà, P., LopezSanchez, J.A., Pagliaro, M.	160	2014	*Chemical Communications*	[84]
	The Bioeconomy to 2030: Designing a policy agenda	Organisation for Economic Cooperation and Development (OECD)	156	2009	_	[85]
Circular Bioeconomy	Waste biorefinery models towards sustainable circular bioeconomy: Critical review and future perspectives	Venkata Mohan, S., Nikhil, G.N., Chiranjeevi, P., (...), Kumar, A.N., Sarkar, O.	259	2016	*Bioresource Technology*	[35]
	Food waste biorefinery: Sustainable strategy for circular bioeconomy	Dahiya, S., Kumar, A.N., Shanthi Sravan, J., (...), Sarkar, O., Mohan, S.V.	113	2018	*Bioresource Technology*	[86]
	A Circular Bioeconomy with Biobased Products from CO2 Sequestration.	VenkataMohan, S., Modestra, J.A., Amulya, K., Butti, S.K., Velvizhi, G.	99	2016	*Trends in Biotechnology*	[34]
	A critical review of organic manure biorefinery models toward sustainable circular bioeconomy: Technological challenges, advancements, innovations, and future perspectives	Awasthi, M.K., Sarsaiya, S., Wainaina, S., (...), Jain, A., Taherzadeh, M.J.	31	2019	*Renewable and Sustainable Energy Reviews*	[87]
	Green Bioplastics as Part of a Circular Bioeconomy	Karan, H., Funk, C., Grabert, M., Oey, M., Hankamer, B.	28	2019	*Trends in Plant Science*	[88]

TA: thematic area; A: article name; AN: name of authors; TC: total number of citations; Y: year of publication of the article; J: name of journals; R: reference.

**Table 4 ijerph-17-09549-t004:** Authors with the most published articles from 2003 through July 2020.

Autores	A	MEPC			TC	Institution	Country	1st A	Last A	References
		CE	BE	CB						
Birch, K	13		X		566	York University	Canada	2009	2019	[89,90,91]
Toppinen, A	12		X	X	413	Helsingin Yliopisto-Helsinki Institute of Sustainability Science	Finland	2014	2020	[95,96]
Pagliaro, M	12	X	X*		224	Istituto Per Lo Studio Dei MaterialiNanostrutturati	Italy	2014	2020	[84,97]
Zabaniotou, A	12	X	X	X	200	Aristotle University of Thessaloniki	Greece	2015	2020	[98,99]
Geng, Y	11	X			1619	China University of Mining and Technology	China	2008	2018	[92,93,94]
Bröring, S	11		X		120	Universität Bonn	Germany	2015	2020	[100,101]
Blumberga, D	8	X	X*		5	Riga Technical University	Latvia	2019	2019	[102,103]
Charnley, F	8	X			177	University of Exeter	United Kingdom	2017	2019	[104,105]
Molina-Moreno, V	8	X			182	Universidad de Granada	Spain	2016	2019	[106,107]
Ciriminna, R	8	X	X *		191	Istituto Per Lo Studio Dei MaterialiNanostrutturati	Italy	2014	2020	[84,97]

A: number of articles; TC: number of citations for all articles; References: Reference of the first and last article and the most cited in some cases; * Indicates most publications are about that CEPM.

**Table 5 ijerph-17-09549-t005:** Percentage of articles by country and by CEPM strategy.

Country	% Articles	CEPM			CES		BE and/or CB		CE Strategy Name	BE Strategy Name	References
		CE	BE	CB	Yes	No	Yes	No			
United Kingdom	15%	79%	20%	1%	X		X		Making Things Last. A Circular Economy Strategy for Scotland (2016) London’s circular economy route map (2017)	Growing the Bioeconomy. Improving lives and strengthening our economy: A national bioeconomy strategy to 2030 (2018)	[109,110,111]
Italy	13%	77%	21%	2%	X		X		Towards a Model of Circular Economy for Italy Overview and Strategic Framework (2017)	BIT II Bioeconomy in Italy. A new bioeconomy strategy for a sustainable Italy (2019)	[112,113]
Germany	12%	43%	56%	1%	X		X		German Resource Efficiency Programme II Programme for the sustainable use and conservation of natural resources (2016) Pathways towards a German Circular Economy Lessons from European Strategies—Preliminary Study (2019)	Nationale Bioökonomiestrategie (2020)	[114,115,116,117]
China	12%	91%	8%	1%	X				Circular Economy Promotion Law of the People’s Republic of China (2009) The 13th Five-Year Plan for Economic and Social Development of the People’s Republic of China (2016–2020) (2016)	13th FYP for Science, Technology and Innovation (2016). 13th FY Development Plan for Strategic Emerging Industries (2016) 13th FYP on Bioindustry Development (2016)	[29,56,118,119,120]
United States	9%	55%	44%	1%			X		-	The Bioeconomy Initiative: Implementing Framework (2019)	[29,121,122]
Spain	8%	80%	20%	1%	X		X		España Circular 2030. Circular Economy. Spanish Strategy (2020)	Spanish Bioeconomy Strategy: Horizon 2030 (2015). Estrategia Española de Bioeconomía. Horizonte 2030. Plan de actuación 2018.	[123,124,125]
Netherlands	6%	75%	25%	1%	X		X		A Circular Economy in the Netherlands by 2050 (2016)	The position of the bioeconomy in the Netherlands (2018)	[126,127]
Finland	5%	48%	50%	4%	X		X		Leading the cycle. Finnish road map to a circular economy 2016–2025 (2016)	The Finnish Bioeconomy Strategy (2014)	[128,129]
France	5%	68%	32%	0%	X		X		The anti-waste law in the daily lives of the French people, what does that mean in practice? Anti-waste law for a circular economy (2020)	A Bioeconomy Strategy for France (2017). A Bioeconomy Strategy for France 2018–2020 Action Plan (2018)	[130,131,132]
Sweden	5%	62%	32%	6%	X		X		Resource Effectiveness and the Circular Economy (2020)	Swedish Research and Innovation Strategy for a Bio-based Economy (2012)	[133,134]
Total	91%	69%	29%	2%							

**Table 6 ijerph-17-09549-t006:** Main institutions from 2003 through July 2020.

Institution	A	%	CEPM			C	IT		(TC) [Reference]
			CE	BE	CB		U	PRC	
Bucharest University of Economic Studies	34	2%	23	11	0	Romania	X		(116) [146]
Delft University of Technology	29	1%	26	3	0	Netherlands	X		(2019) [77]
Lunds Universitet	29	1%	16	11	2	Sweden	X		(1161) [82]
Università degli Studi di Catania	28	1%	20	8	0	Italy	X		(184) [147]
Universität Hohenheim	26	1%	0	26	0	Germany	X		(376) [147]
Wageningen University & Research	25	1%	8	17	0	Netherlands	X		(459) [148]
European Commission Joint Research Centre	25	1%	7	18	0	Belgium		X	(725) [81]
Chinese Academy of Sciences	22	1%	21	0	1	China		X	(1626) [93]
Consiglio Nazionale delle Ricerche	21	1%	9	11	1	Italy		X	(250) [84]
The University of Manchester	20	1%	20	0	0	United Kingdom	X		(311) [149]
Total	259	12%	150 (58%)	105 (41%)	4 (2%)				

A: number of articles; C: country; IT: institution type; U: university; PRC: public research center; TC: total number of citations; Reference: most cited article.

**Table 7 ijerph-17-09549-t007:** Journals with the largest number of publications from 2003 through July 2020.

Journal	A	CEPM			Journal H index	SJR	C	(TC) [Reference]
		CE	BE	CB				
*Journal of Cleaner Production*	222	186	34	3	173	1.886(Q1)	Netherlands	(9650) [17]
*Sustainability*	187	132	54	1	68	0.58 (Q2)	Switzerland	(2946) [151]
*Resources Conservation and Recycling*	87	87	0	0	119	2.22 (Q1)	Netherlands	(2728) [79]
*Amfiteatru Economic*	35	12	23	0	18	0.28 (Q2)	Romania	(81) [146]
*Bioresource Technology*	32	15	3	14	273	2.43 (Q1)	Netherlands	(915) [35]
*Journal of Industrial Ecology*	30	29	1	0	95	1.81 (Q1)	United States	(1949) [153]
*Waste Management*	25	25	0	0	145	1.63 (Q1)	United Kingdom	(390) [154]
*Science of the Total Environment*	25	22	2	1	224	1.66 (Q1)	Netherlands	(248) [155]
*Industrial Biotechnology*	22	2	19	1	30	0.31 (Q3)	United States	(129) [156]
*Biofuels Bioproducts and Biorefining*	22	1	20	1	78	1.14 (Q1)	United Kingdom	(223) [157]
Total	687	511	156	21				
	31%	74%	23%	3%				

A: number of articles; SJR: SCImago Journal Rank; C: country; TC: total number of citations; Reference: most cited article.

**Table 8 ijerph-17-09549-t008:** Main keywords from 2003 through July 2020.

Sustainability	Recycling	Sustainable Development	Waste Management	Industrial Ecology
223 (10%)	107 (5%)	91 (4%)	69 (3%)	55 (3%)
Resource efficiency	Life cycle assessment	Waste	Bioenergy	China
48 (2%)	43 (2%)	39 (2%)	38 (2%)	35 (2%)
Innovation	Industrial symbiosis	Reuse	Biotechnology	Biomass
32 (1%)	31 (1%)	31 (1%)	31 (1%)	29 (1%)
Environment	Remanufacturing	European Union	Renewable energy	Agriculture
20 (1%)	20 (1%)	19 (1%)	19 (1%)	18 (1%)

**Table 9 ijerph-17-09549-t009:** Main works about agriculture according to citations from 2008 through July 2020.

Year	Document title	TC	CEPM		R	Year	Document title	TC	CEPM		Reference
			CE	BE					CE	BE	
2013	Divergent Paradigms of European Agro-Food Innovation: The Knowledge-Based Bio-Economy (KBBE) as an R&D Agenda	87		X	[165]	2015	Boosting circular economy and closing the loop in agriculture: Case study of a small-scale pyrolysis-biochar based system integrated in an olive farm in symbiosis with an olive mill.	30	X		[98]
2012	EU agri-innovation policy: Two contending visions of the bio-economy	69		X	[166]	2019	Contribution to Circular Economy options of mixed agricultural wastes management: Coupling anaerobic digestion with Gasification for enhanced energy and material recovery.	27	X		[45]
2009	From the petro-economy to the bioeconomy: Integrating bioenergy production with agricultural demands	33		X	[167]	2013	Twenty-first century bioeconomy: Global challenges of biological knowledge for health and agriculture.	24		X	[168]
2017	Design of marine macroalgae photobioreactor integrated into building to support seagriculture for biorefinery and bioeconomy.	30		X	[169]	2016	The seven challenges for transitioning into a bio-based circular economy in the agri-food sector.	19	X		[170]
2016	Towards a Circular Economy in Australian Agri-food Industry: An Application of Input-Output Oriented Approaches for Analyzing Resource Efficiency and Competitiveness Potential	30	X		[171]	2018	An efficient agro-industrial complex in Almería (Spain): Towards an integrated and sustainable bioeconomy model.	15		X	[172]

**Table 10 ijerph-17-09549-t010:** Research about AWB.

Title	Year	CEPM		Reference	Title	Year	CEPM		Reference
		CE	BE/CB				CE	BE/CB	
A novel compost for rice cultivation developed by rice industrial by-products to serve circular economy	2019	X		[174]	Promoting Circular Economy Through Sustainable Agriculture in Hidalgo: Recycling of Agro-Industrial Waste for Production of High Nutritional Native Mushrooms	2019	X		[48]
A spatial approach to bioeconomy: Quantifying the residual biomass potential in the EU-27	2019		X	[43]	Refining biomass residues for sustainable energy and bio-products: An assessment of technology, its importance, and strategic applications in circular bio-economy	2020		X	[37]
Agriculture waste valorisation as a source of antioxidant phenolic compounds within a circular and sustainable bioeconomy	2020		X	[47]	Role of biogenic waste and residues as an important building block towards a successful energy transition and future bioeconomy–Results of a site analysis	2020		X	[31]
Are Primary Agricultural Residues Promising Feedstock for the European Bioeconomy?	2017		X	[15]	Sugarcane: A potential agricultural crop for bioeconomy through biorefinery	2017		X	[13]
Assessment of agroforestry residue potentials for the bioeconomy in the European Union	2018		X	[46]	Ten-year legacy of organic carbon in non-agricultural (brownfield) soils restored using green waste compost exceeds 4 per mille per annum: Benefits and trade-offs of a circular economy approach	2019	X		[175]
Bioeconomy and the production of novel food products from agro-industrial wastes and residues under the context of food neophobia	2018		X	[176]	The bioeconomy of microalgal heterotrophic bioreactors applied to agroindustrial wastewater treatment	2017		X	[177]
Camelina and crambe oil crops for bioeconomy-straw utilisation for energy	2020		X	[178]	The circular economy of agro and post-consumer residues as raw materials for sustainable ceramics	2020	X		[50]
Cellulolytic enzyme production from agricultural residues for biofuel purpose on circular economy approach	2019	X		[179]	The management of agricultural waste biomass in the framework of circular economy and bioeconomy: An opportunity for greenhouse agriculture in Southeast Spain	2020	X	X	[7]
Co-digestion of by-products and agricultural residues: A bioeconomy perspective for a Mediterranean feedstock mixture	2020		X	[180]	The potential of plantain residues for the Ghanaian bioeconomy-assessing the current fiber value web	2018		X	[181]
Contribution to Circular Economy options of mixed agricultural waste management: Coupling anaerobic digestion with gasification for enhanced energy and material recovery	2019	X		[45]	Tomato’s Green Gold: Bioeconomy Potential of Residual Tomato Leaf Biomass as a Novel Source for the Secondary Metabolite Rutin	2019		X	[182]
Fruit waste streams in South Africa and their potential role in developing a bio-economy	2015		X	[173]	Towards circular economy solutions for the management of rice processing residues to bioenergy via gasification	2019	X		[44]
Intermediate pyrolysis of agricultural waste: A decentral approach towards circular economy	2018	X		[183]	Valorising agro-industrial wastes within the circular bioeconomy concept: The case of defatted rice bran with emphasis on bioconversion strategies	2020		X	[184]
Planning the Flows of Residual Biomass Produced by Wineries for Their Valorization in the Framework of a Circular Bioeconomy	2020		X	[185]	Valorization of agricultural waste for biogas-based circular economy in India: A research outlook	2020	X		[186]

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
