# Peer review of "Analysis of the Circular Economic Production Models and Their Approach in Agriculture and Agricultural Waste Biomass Management"

_ijerph, 2020, doi:10.3390/ijerph17249549_

Round 1
Reviewer 1 Report
My considerations are outlined below:
- Line #84: I believe that the new paragraph that starts here breaks the logical conciseness of the former. Merging the referred paragraph ("Many studies...") to the former would improve the readbility, in my opinion;
- The authors should try to reduce the reference of other studies conducted from one or more of the authors. Alltough they apper in the present paper in a reasonable ammount, I would recommend for the authors to consider the replacement of the self-citated references for different ones, as many as possible.
- Line #409: It would be benefitial if the authors clearly describe what software was used to produce the biliometric network presented in the Figures 4 and 5, and other figures presented in the text. Alltought the logo of VOSViewer software is stated in all figures, the explicit description in the text would benefit the paper readbility.
- The authors do not have included any information regarding hydric footprint. I belive that this concern poses an important restriction for many activites (eg: intensive cattle raising, among others) and should be included in the text, at least as an marginal discussion in some extent.
Author Response
Please, see the attachment.

Reviewer 2 Report
The document appears to be interesting but does not delve into the subject at hand.
The methodology does not seem to be presented adequately:
1. There is no explanation why to use the SCOPUS metasearch engine
2. From which period the search for results is carried out.
3. In the search filters it is not detailed if other languages, disciplines, etc. are considered.
The results are not discussed and are presented in a descriptive and quantitative manner.
It may be interesting to carry out a meta-analysis of the information
Author Response
Please, see the attachment.

Reviewer 3 Report
Suggested points:
- Abstract
- The CEPMs follow in the first line where Circular Economic Production Model (line 14)
- From the abstract it is very difficult to identify the novel findings of this analysis
- What is bibliometric analysis?, I am asking as an ordinary reader (e.g. statistical method to analyze articles.., what is the method?)
- Introduction
- Good start of introduction, but difficult to use a single letter as an acronym e.g. B for Bioeconomy
- What are the agricultural waste biomass (AWB), what does show the statistics:- e.g.?
- Table 1 shows CEPMs studies from 2003, but in the table why is only from 2017 to 2020 only?
- In general the introduction needs more elaboration on the objective of this study
- Materials and Methods
- Explain what is VOSviewer software (e.g. A software tool for analyzing and visualizing scientific literature)
- Section 2.2. the main methodological process needs some explanation in the M & Ms. i.e. not only the model flow
- M & Ms needs specific explanation on the method specially on the VOSviewer analysis
- Results and Discussion
- Fig 2. Is the y-aixs title of Articles by year? (should indicate number)
- Please replace the letter B with Bioeconomy, no acronym
- Line 212 CO2
- Tables need more editing
- Table 5 needs more editing
- In Fig 4 & 5 What is the need to have VOSviewer? (Explanation?)
- Edit table 9
- Edit table 10
- References
- Needs reference checking
Author Response
Please, see the attachment.

Round 2
Reviewer 3 Report
Abstract
- agricultural waste biomass (AWB)
Tables-
- Table 5, the title is at the bottom of the table and there is extra page
Fig. 6
- y-axis for % should be (0-100%), the legend % only not enough What %?
Fig. 7
- What is the title of the Y-axis (No. of publication ?)
Fig. 9
- remove the numbers since you have the x-axis
In general, some of the graphs quality is not satisfactory.
Author Response
Please, see the attachment. Thanks!
